

# Effects of thiourea on the skull of *Triturus* newts during ontogeny

Maja Ajduković[1], Tijana Vučić[2] and Milena Cvijanović[1]

[1] Department of Evolutionary Biology, Institute for Biological Research "Siniša Stanković", National Institute of the Republic of Serbia, University of Belgrade, Belgrade, Serbia
[2] Institute of Zoology, Faculty of Biology, University of Belgrade, Belgrade, Serbia

## ABSTRACT

**Background**. In amphibians, thyroid hormone (TH) has a profound role in cranial development, especially in ossification of the late-appearing bones and remodeling of the skull. In the present study, we explored the influence of TH deficiency on bone ossification and resulting skull shape during the ontogeny of *Triturus* newt hybrid larvae obtained from interspecific crosses between *T. ivanbureschi* and *T. macedonicus*.
**Methods**. Larvae were treated with two concentrations of thiourea (an endocrine disruptor that chemically inhibits synthesis of TH) during the midlarval and late larval periods. Morphological differences of the cranium were assessed at the end of the midlarval period (ontogenetic stage 62) and the metamorphic stage after treatment during the late larval period.
**Results**. There was no difference in the ossification level and shape of the skull between the experimental groups (control and two treatment concentrations) at stage 62. During the late larval period and metamorphosis, TH deficit had a significant impact on the level of bone ossification and skull shape with no differences between the two treatment concentrations of thiourea. The most pronounced differences in bone development were: the palatopterygoid failed to disintegrate into the palatal and pterygoid portions, retardation was observed in development of the maxilla, nasal and prefrontal bones and larval organization of the vomer was retained in thiourea-treated larvae.
**Conclusions**. This implies that deficiency of TH caused retardation in development and arrested metamorphic cranium skeletal reorganization, which resulted in divergent cranial shape compared to the control group. Our results confirmed that skull remodeling and ossification of late-appearing bones is TH–dependent, as in other studied Urodela species. Also, our results indicate that TH plays an important role in the establishment of skull shape during the ontogeny of *Triturus* newts, especially during the late larval period and metamorphosis, when TH concentrations reach their maximum.

## INTRODUCTION

Thyroid hormones (thyroxine T4 and triiodothyronine T3), produced by the thyroid gland, play an important and widespread role in skeletal differentiation and morphogenesis in vertebrates (*Hanken & Summers, 1988*; *Rose & Reiss, 1993*; *Smirnov & Vassilieva, 2003a*). In

Corresponding author
Maja Ajduković,
maja.ajdukovic@ibiss.bg.ac.rs

amphibians, thyroid hormone (TH) has a crucial role in metamorphosis, which represents dramatic structural and functional changes of larval tissue to the juvenile form (*Kollros, 1961*; *Dodd & Dodd, 1976*). It is well known that TH synthesis and release to the bloodstream increases towards metamorphosis, reaches a maximum at the climax stage and decreases after metamorphosis (*Regard, Taurog & Nakashima, 1978*; *Larras-Regard, Taurog & Dorris, 1981*; *White & Nicoll, 1981*; *Alberch, Gale & Larsen, 1986*; *Norman, Carr & Norris, 1987*; *Sternberg et al., 2011*). Also, TH plays a profound role in the skeletal development of amphibians, especially in remodeling of the skull, which undergoes abrupt changes during metamorphosis (*Rose, 2003*; *Smirnov & Vassilieva, 2003a*). During amphibian ontogeny, the timing of cranial bone ossification shows species-specific characteristics and is TH-mediated (*Rose & Reiss, 1993*; *Smirnov & Vassilieva, 2003b*). Ossification sequences of cranial bones are therefore determined by TH concentration and by the sensitivity of osteogenic sites to TH (*Smirnov & Vassilieva, 2003b*). According to *Smirnov & Vassilieva (2003a)*, bones appearing early in the skull (coronoid, palatine, dentary, vomer, premaxilla, squamosal, angular, pterygoid, parasphenoid), ones appearing later (parietal and frontal), and midlarval bones (exoccipital, orbitosphenoid, quadrate and prootic) are not TH-inducible. However, late-appearing bones such as the maxilla, prefrontal and nasal, as well as metamorphic skull remodeling, are TH-inducible (*Smirnov & Vassilieva, 2003a*; *Smirnov, Vassilieva & Merkulova, 2011*).

Different studies investigated the role of TH in cranial development by treating larvae not only with exogenous TH but also with the goitrogenic substance thiourea, a chemical inhibitor of thyroid synthesis (*Smirnov & Vassilieva, 2003a*; *Smirnov & Vassilieva, 2003b*; *Smirnov, Vassilieva & Merkulova, 2011*). Thiourea (CAS Number: 62-56-6) is well established in treating hyperactivity of the thyroid gland (hyperthyroidism) and causes a decrease in the level of TH (*Gobdon, Goldsmith & Charipper, 1943*; *Wheeler, 1953*; *Chakraborty et al., 2018*). This anti-thyroid agent inhibits TH synthesis by the thyroid gland but does not block it completely, like the action of thyroidectomy, which represents surgical removal of the thyroid gland (*Brown, 1997*). The mechanism of anti-thyroid agents is realized via action on inhibition of thyroid peroxidase (TPO) (*Engler, Taurog & Nakashima, 1982*; *Wegner, Browne & Dix, 2016*), the enzyme responsible for oxidation of iodide in the biosynthesis of thyroid hormones. In the presence of iodine, thiourea competes with tyrosine residues of thyroglobulin (the precursor of thyroid hormones) for oxidized iodine, thus diverting oxidized iodine away from thyroglobulin (*Sheikh & McGregor, 1997*). At last, anti-thyroid drugs themselves are oxidized and degraded. The use of thiourea, a well-known endocrine disruptor, revealed a strong inhibitory effect on thyroid hormone synthesis in amphibians (*Takamura et al., 2020*). Prolonged treatment with thiourea results in a hypertrophic and hyperplastic thyroid gland due to the pituitary gland's overproducing thyroid-stimulating hormone (TSH) in response to the drop in the level of TH in the blood. An extensive literature is available for all tetrapod groups concerning the effects of thiourea on morphology and physiology of the thyroid gland. Among amphibians, the majority of studies treated the ability of thiourea to inhibit metamorphosis in tadpoles (*Gobdon, Goldsmith & Charipper, 1943*; *Lynn, 1948*; *Steinmetz, 1950*; *Krishnapriya et al., 2014*) and its influence on cranial osteology of larvae and newly

PeerJ ___________________________________________

metamorphosed newts under conditions of an artificially caused deficit of TH (*Smirnov & Vassilieva, 2003a*; *Smirnov & Vassilieva, 2003b*; *Smirnov & Vassilieva, 2005*; *Smirnov, Vassilieva & Merkulova, 2011*). Despite numerous studies dealing with the influence of TH on induction of the appearance of cranial bones and ossification in Urodela (*Rose, 1995a*; *Rose, 1995b*; *Smirnov & Vassilieva, 2003a*; *Smirnov & Vassilieva, 2003b*; *Smirnov, Vassilieva & Merkulova, 2011*), there have been no direct studies on the mediation of skull shape by TH in Urodela.

The order Urodela is one of the major extant orders of the class Amphibia. It includes salamanders and newts which, are characterized by slender bodies, blunt snouts, short limbs and the presence of a tail in both larvae and adults. The genus *Triturus* forms a well-supported monophyletic clade of newts within the family Salamandridae (*Steinfartz et al., 2007*; *Wielstra & Arntzen, 2011*). According to current taxonomy, the genus *Triturus* consists of nine species: two marbled (*T. marmoratus* and *T. pygmaeus*) and seven crested (*T. karelinii*, *T. ivanbureschi*, *T. anatolicus*, *T. macedonicus*, *T. carnifex*, *T. cristatus*, and *T. dobrogicus*) newt species (*Wielstra et al., 2019*).

The present study aimed to investigate whether altered levels of TH during larval ontogeny have an impact on cranial morphology in *Triturus* newt hybrid larvae and juveniles. We set up two experiments. In the first experiment, larvae were treated with a goitrogenic substance during the midlarval period and examined at the end of this period, at the ontogenetic stage 62. In the second experiment, larvae were treated during the late larval period and metamorphosis and examined at the metamorphic stage. At these stages, we obtained the level of bone ossification, estimated skull shape and compared them in two thiourea-treated groups and control. We expected that a TH deficit would have a major impact on cranial morphology during the late larval period and metamorphosis, when thyroid hormones reach their maximum level and have a crucial role in the development of late-appearing bones. Accordingly, we expected that differences in cranial shape between treated and control groups will be pronounced at the metamorphic stage.

# MATERIALS & METHODS

## Animal housing

We used hybrids obtained from crosses between *T. ivanbureschi* and *T. macedonicus*. These two species hybridize at their contact zones in a natural populations (*Arntzen, Wielstra & Wallis, 2014*; *Wielstra et al., 2017*). In natural unimodal hybrid populations, hybrids cannot be distinguished based on external morphology from parental species (*Arntzen et al., 2018*). Additionally, it has been shown in the laboratory that values of tail shape, size and colouration patterns (*Vučić et al., 2018*), as well as general head shape during larval development and after metamorphosis (*Vučić et al., 2019*), are intermediate between hybrids and the parental species during ontogeny (*Vučić et al., 2019*). Taking into account the aforementioned, we decided to use hybrid larvae (obtained from experimental crosses) to explore possible differences in cranial morphology due to different TH levels during larval development, since we were not interested in interspecific divergence at this point.

After mating in common containers, gravid females were transferred separately to aquaria containing plastic strips for egg deposition. Eggs were collected daily and kept

submerged in dechlorinated tap water in plastic Petri dishes until hatching. After that, hatchlings were transferred to a small plastic cup with dechlorinated tap water, where they developed in laboratory conditions under a natural day-night light regime at air temperatures of 18° ±1 °C and a water temperature of 16° ±1 °C. Larvae were fed *ad libitum* with *Artemia* sp. at earlier developmental stages and with *Tubifex* sp. at later stages. The stage of larvae was determined according to the developmental staging table for crested newts (*Glücksohn, 1931*), which is based on limb development and digit formation (described only by external features). A degree of limb differentiation appears to be suitable for ontogenetic staging under different TH regimes because limb development is TH independent. For larval developmental stage 50, the basic features are formation of the fourth digit in forelimbs and initiation of hindlimb bud formation. Developmental stage 62 is characterized by fully developed limbs, formed external gills and the appearance of larval skin pigmentation, unlike metamorphic individuals, which are characterized by the resorption of external gills and closure of gill slits. Also, dramatic changes are notable in the skin, especially in its pigmentation and thickness, in metamorphic individuals. A brief overview of developmental stages from hatching until metamorphosis with graphic presentation of the analyzed stages is given in Table S1.

## Experimental settings

A detailed protocol for the experimental design and an analysis plan suiting the research questions posed were prepared in advance. The first experiment started at larval stage 50 and lasted throughout the middle larval period until the larvae reached stage 62 (*Glücksohn, 1931*; see Table S1). When the larvae ($N = 26$) reached stage 50, they were transferred to 2-L plastic containers half-filled with different media depending on the experimental group. Three groups were defined: (1) control (dechlorinated tap water), (2) low thiourea concentration (0.05% solution of thiourea) and (3) high thiourea concentration (0.1% solution of thiourea). Thiourea (p.a. $\geq$99.0%; Sigma, St. Louis, MO, USA) solutions (low and high concentrations) were prepared in dechlorinated tap water. These two concentrations of thiourea were established in previous work on salamanders (*Wheeler, 1953*). The duration of exposure to thiourea was approximately three months for each experiment. To minimize confounding effects, the larvae of the control and two treatment groups were kept under the same laboratory conditions at the same density (five larvae per container). The medium was changed every second day. When larvae reached stage 62 (*Glücksohn, 1931*; Table S1), they were euthanized with ethyl 3-aminobenzoate methanesulfonate better known as MS 222 (CAS Number: 886-86-2; Sigma, St. Louis, MO, USA), and preserved whole in 96% ethanol.

The second experiment started at stage 62 and lasted throughout the late larval period until larvae ($N = 49$) from the control group complete metamorphosis. Because thiourea inhibits TH synthesis and therefore inhibits the ability of larvae to metamorphose, sampling of larvae was based on normally metamorphosing individuals in the control group. Metamorphosed individuals are characterized by the resorption of external gills and closure of gill slits. Also, some differences in the skin are notable, especially in pigmentation and thickness (*Glücksohn, 1931*; Table S1). Photos of cleared and stained skulls of analyzed

larvae at the beginning and end of the second experiment are given as Supplemental Fig. 1. The same experimental design, viz., control group and two concentrations of thiourea [low (0.05%) and high (0.1%)], was used for the second experiment. For the second experiment, there were three larvae per container to avoid potential high larval density caused by the increase in larval body mass and size, which invariably leads to some detrimental effects, such as reduction in larval growth or size at metamorphosis (*Altwegg, 2003*). Larvae were reared in 2-L plastic containers half-filled with thiourea media, while the control group was raised in dechlorinated tap water until the end of metamorphosis, when all animals were sampled, euthanized and preserved whole in 96% ethanol. There were no differences in dissolved oxygen, pH and temperature between the control group and two thiourea concentrations (0.05 and 0.1%). Also, the mortality rate and other growth effects were similar in the control group and at two thiourea concentrations. All animals used in both experiments were included in the analysis and randomly allocated to treatment and control groups.

All analysed larvae were prepared as skeletal whole-mounts. Materials were cleared with trypsin and 1% KOH and double-stained for cartilage and bone with Alcian blue and alizarin red S, respectively (*Dingerkus & Uhler, 1977*). Whole-mounts of larvae were stored in the batrachological collection of the ''Siniša Stanković'' Institute for Biological Research.

Collection of adult animals from natural populations was approved by the Ministry of Energy, Development and Environmental Protection of the Republic of Serbia (permit no. 353-01-75/2014-08) and the Environmental Protection Agency of Montenegro (permit no. UPI-328/4). The experiment was approved by the Ethical Committee of the ''Siniša Stanković'' Institute for Biological Research (decision no. 02-07/19). All experimental animals were treated in compliance with a European directive (2010/63/EU) on the protection of animals used for experimental and other scientific purposes.

## Data collection

We used the minimal sample size per group (an individual larva was treated as an experimental unit) in order to be able to perform a valid statistical analysis. The cleared and stained skulls of analyzed larvae for stage 62 ($N = 26$) and the metamorphic stage ($N = 49$) were photographed against a scale bar (10 mm) in Petri dishes positioned in the centre of the optical field using a Nikon Digital Sight Fi2 camera attached to a Nikon SMZ800 stereo zoom microscope. Skull photographs of high resolution were used for scoring the level of cranial element ossification and skull shape.

We scored the presence and ossification level of skeletal elements in the skull at two developmental stages for both the dorsal and the ventral side. Scores were in the range of 0 to 2, where 0 represents the absence of bone, 1 stands for partially ossified bones and 2 represents fully ossified bones. At stage 62, ossification scores were obtained from 10 larvae not treated (control), 10 larvae treated with 0.05% thiourea (low concentration) and six larvae treated with 0.1% thiourea (high concentration). For that developmental stage, we scored the exoccipital, frontal, parietal, premaxilla and squamosal bones on the dorsal skull side, and the palatopterygoid, parasphenoid and vomer bones on the ventral side of the skull.

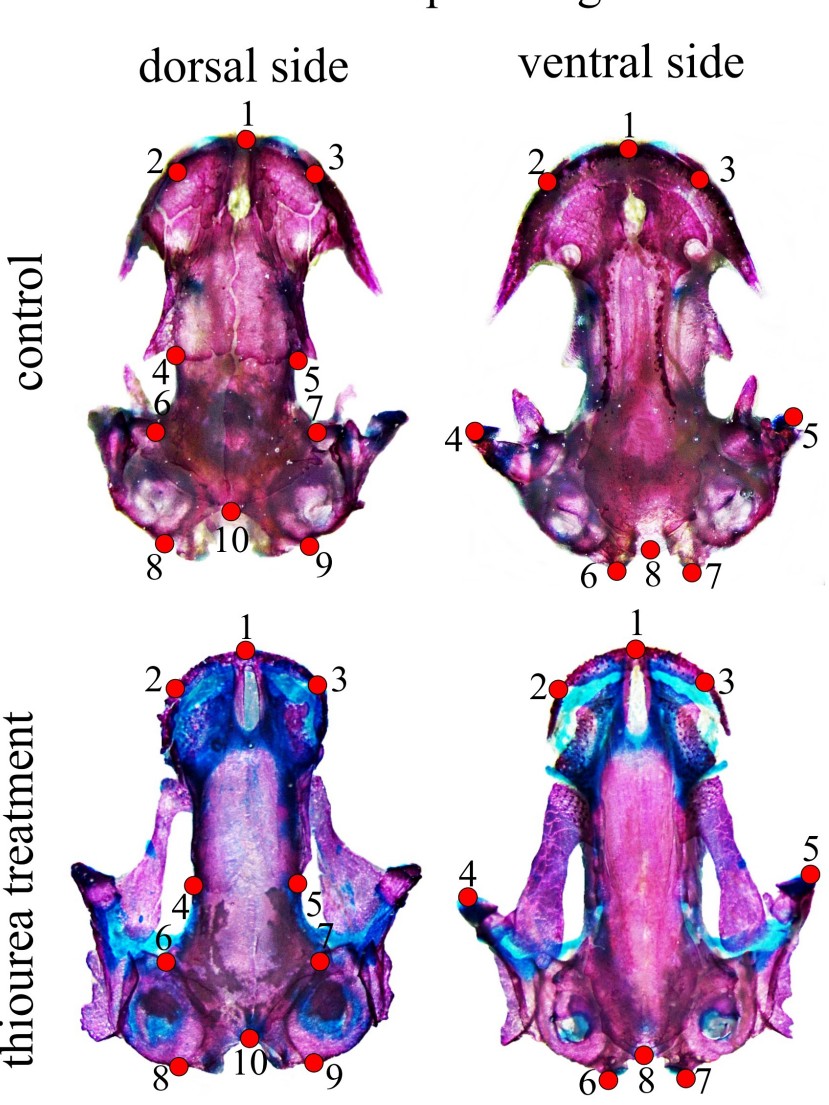

**Figure 1** **Configuration of two-dimensional landmarks for the dorsal and ventral skull at metamorphic stages.**

At the metamorphic stage, ossification scores were obtained from 14 individuals in the control group, 19 individuals treated with 0.05% thiourea (low concentration) and 16 individuals treated with 0.1% thiourea (high concentration). For that developmental stage, we scored the frontal, maxilla, nasal, parietal, prefrontal and squamosal bones on the dorsal skull side, and the palatopterygoid, pterygoid and vomer bones on the ventral side of the skull.

At stage 62, a configuration of eight two-dimensional landmarks for both the dorsal and the ventral cranium was used to describe larval skull shape. For the metamorphic stage, 10 two-dimensional landmarks for the dorsal and eight two-dimensional landmarks

**Table 1** Differences between groups in the frequencies of ossification level states at ontogenetic stage 62 and the metamorphic stage.

| Stage 62 | G test | p | Metamorphic | G test | p |
|---|---|---|---|---|---|
| frontal | 0.35 | 0.841 | nasal | 51.81 | <0.001 |
| parietal | 0.35 | 0.841 | prefrontal | 50.87 | <0.001 |
| squamosal | 0.91 | 0.631 | maxilla | 39.61 | <0.001 |
| exoccipital | 3.02 | 0.221 | vomer | 55.63 | <0.001 |
| palatopterygoid | 3.36 | 0.187 | palatopterygoid | 55.63 | <0.001 |
| parasphenoid | 4.27 | 0.118 | pterygoid | 55.63 | <0.001 |

for the ventral cranium were used (Fig. 1). A brief anatomical description of the position of cranium landmarks for both stages is given in Table S2. All landmarks were digitized by the same person (MA) using TpsDig software (*Rohlf, 2005*). Centroid size (CS) as a measure of size was calculated in the CordGen program from the IMP package (*Sheets, 2000*). Generalized Procrustes analysis (GPA) was employed to obtain shape coordinates (Procrustes coordinates) (*Rohlf & Slice, 1990*; *Dryden & Mardia, 1998*) using the Morpho J software package (*Klingenberg, 2011*). Procrustes coordinates were used in further analysis of shape changes. The data used for statistical analysis are accessible as supplementary material.

## Statistical analyses

The G test was used to compare the frequencies of observed character states (different levels of ossification) of analyzed bones. William's correction for small sample size was applied. Multiple correspondence analysis (MCA) was performed to explore the potential influence of thiourea on bone ossification and development.

We used PCA to explore and visualize variation in skull shape. To analyze possible differences in skull shape, Procrustes distances between mean shapes of the three analyzed groups were calculated. Permutation tests with 1000 iterations were performed to obtain the pairwise difference between the mean values of cranium shape.

The Statistica® 10.0 computer package (StatSoft, Inc., Tulsa, OK) was used in performing MCA. The G test was performed using Microsoft Excel. Landmark-based analysis, and graphic visualizations were run in the Morpho J software package (*Klingenberg, 2011*).

## RESULTS

### Stage 62

At ontogenetic stage 62, all analyzed bones were present in all examined individuals. The premaxilla and vomer were completely ossified in all three groups. The other analyzed bones (exoccipital, frontal, palatopterygoid, parasphenoid, parietal and squamosal) were partially ossified with subtle differences in the level of ossification among individuals from all three analyzed groups with no statistically significant difference among groups (G test, $p > 0.1$ in all comparisons; see Table 1).

Permutation tests showed that there was no statistically significant difference in cranium shape between the tested groups for either the dorsal or the ventral cranium (Table 2).
**Table 2** Procrustes distances (PD) and *p*-values from permutation tests (with 1000 permutation rounds).

| Compared groups | Stage 62 | | | | Metamorphosis | | | |
|---|---|---|---|---|---|---|---|---|
| | Dorsal | | Ventral | | Dorsal | | Ventral | |
| | PD | *p* | PD | *p* | PD | *p* | PD | *p* |
| control–low | 0.0188 | 0.242 | 0.0129 | 0.685 | 0.0473 | <0.001 | 0.0463 | <0.001 |
| control–high | 0.0265 | 0.131 | 0.0239 | 0.205 | 0.0584 | <0.001 | 0.0635 | <0.001 |
| low–high | 0.016 | 0.263 | 0.0223 | 0.111 | 0.0176 | 0.206 | 0.0178 | 0.354 |

**Table 3** Between-group comparisons of the frequencies of ossification level states at the metamorphic stage. The vomer, palatopterygoid and pterygoid bones have the same level of ossification in high and low thiourea-treated groups.

| bones | control–high | | control–low | | low–high | |
|---|---|---|---|---|---|---|
| | G test | *p* | G test | *p* | G test | *p* |
| nasal | 37.91 | <0.001 | 35.29 | <0.001 | 2.77 | 0.251 |
| prefrontal | 38.29 | <0.001 | 35.29 | <0.001 | 1.95 | 0.378 |
| maxilla | 32.44 | <0.001 | 35.94 | <0.001 | 1.85 | 0.396 |
| vomer | 39.46 | <0.001 | 39.46 | <0.001 | – | – |
| palatopterygoid | 39.46 | <0.001 | 42.91 | <0.001 | – | – |
| pterygoid | 39.46 | <0.001 | 42.91 | <0.001 | – | – |

## Metamorphic stage

At this ontogenetic stage, the frontal, parietal and squamosal bones had the same level of ossification in all three groups. The other analyzed bones (nasal, maxilla, prefrontal, palatopterygoid, pterygoid and vomer) differed in frequencies of the observed level of ossification states (G test, $p < 0.001$; see Table 1). Between-group comparisons showed that the control group differed from both thiourea groups (G test, $p < 0.001$ for the analyzed bones; see Table 3), while there were no differences between the high and low thiourea concentration groups (G test, $p > 0.2$ for the analyzed bones; see Table 3).

At the metamorphic stage, the first axes obtained by MCA explained 65.10% of the total inertia describing the amount of variation (Fig. 2). On the dorsal side, the maxilla, nasal and prefrontal bones were fully developed in the control group, while larvae subjected to treatment with low and high concentrations of thiourea were mainly characterized by retarded development or even absence of the aforementioned bones. On the ventral side of the skull, control individuals were characterized by a fully developed vomer, unlike individuals from the two thiourea-treated groups, which had vomers with numerous resorption pits, the presence of which is characteristic of the larval skull. Complete disintegration of the palatopterygoid into palatal and pterygoid bones was present in the control group, unlike in the thiourea-treated individuals, where the palatopterygoid showed no signs of disintegration.

Permutation tests showed that there was a statistically significant difference in skull shape between larvae from the control group and the two thiourea-treated groups, but there was no statistically significant difference between the groups receiving low and high

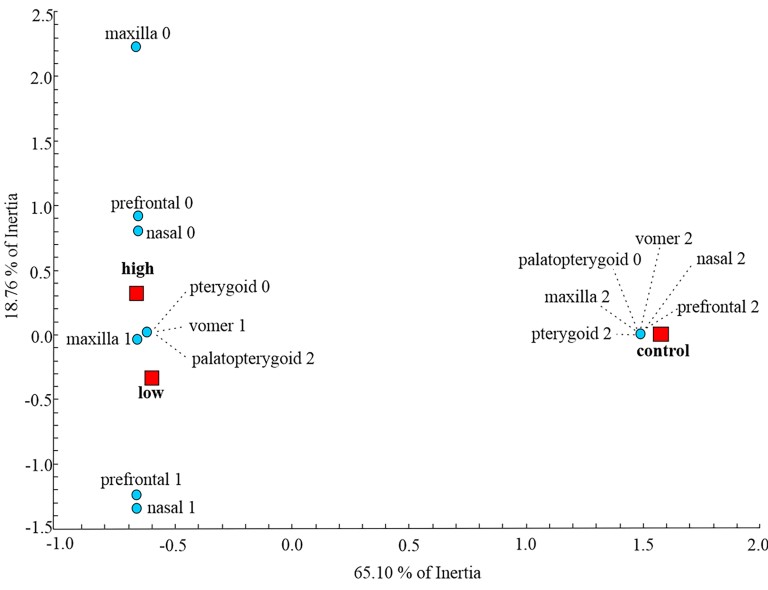

**Figure 2** **Multiple Correspondence Analysis of skull ossification.** Graphic presentation of the positions of control and two thiourea treatment groups (red squares) relative to the first two axes obtained by MCA on skull character state scores (blue circles) for the metamorphic stage. The inertia represents the amount of variation of skull character state scores described by each axis.

concentrations of thiourea (Table 2). For the dorsal cranium, the two thiourea groups separated from the control group along the PC1 axis which described 35.15% of the total variance. Thiourea-treated individuals retained late larval shape with a narrow cranium, especially in the middle part (the suture between the frontal and parietal bones; landmarks 4 and 5) and the otico-occipital region (landmarks 6, 7, 8, 9 and 10). The anterior part of the skull (on the level of the premaxilla; landmarks 1, 2 and 3) is slightly more slender in thiourea-treated individuals compared to those of the control group (Fig. 3). For the ventral cranium, the PC1 axes described 60.73% of total variation and differences between the two thiourea groups and the control group. The most pronounced characteristic is that thiourea-treated individuals possess an elongated quadratum (landmarks 4 and 5). In the anterior part of the skull (on the level of the premaxilla; landmarks 1, 2 and 3) and in the cranium base (landmarks 6, 7 and 8), differences between thiourea-treated individuals and the control group are less pronounced (Fig. 3).

## DISCUSSION

In general, thyroid hormone (TH) plays an important role in many physiological processes involved in growth, development and behaviour, and it is essential for normal cranial development. Regulation of TH levels is maintained by a negative feedback loop involving the hypothalamus-pituitary-thyroid (HPT) axis (*Kim & Mohan, 2013*; *Bassett & Williams, 2016*; *Ortiga-Carvalho et al., 2016*). Thyroid actions are exerted through TH receptors (TR-alpha and TR-beta), which are members of the nuclear receptor (NR) superfamily and are ligand-inducible transcription factors (*Terrien & Prunet, 2013*). In amphibians,

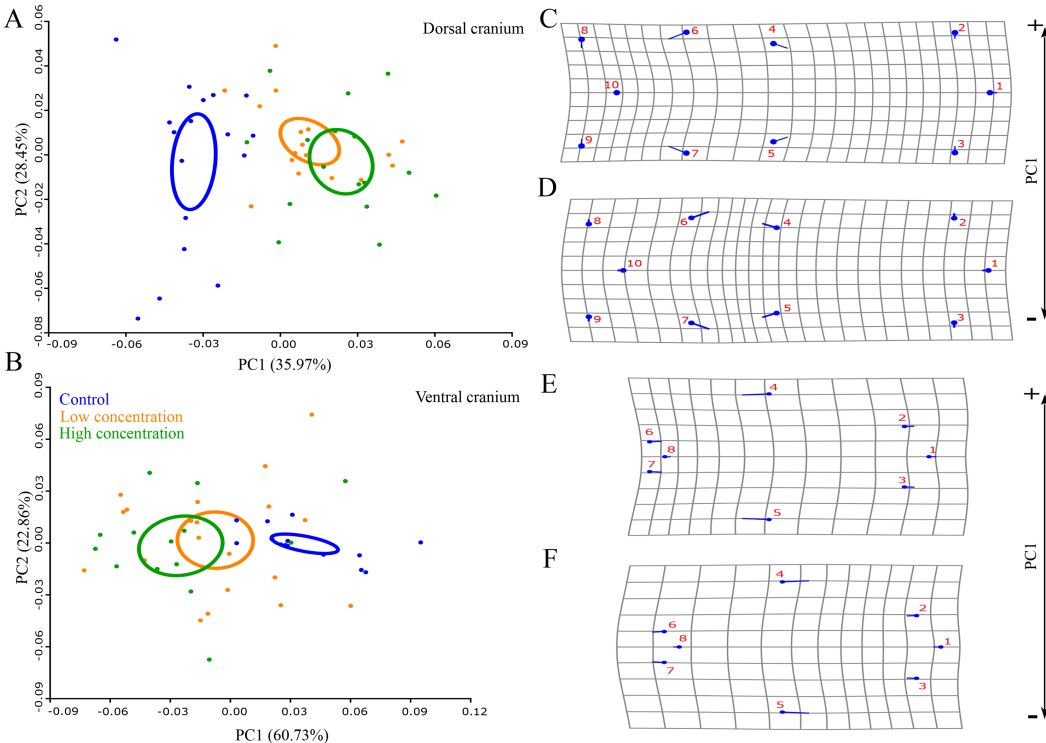

**Figure 3 Principal Component Analysis and shape changes of the skull.** Position of control and two thiourea treatment groups (A and B) with deformation grids as an illustration of shape changes (C–F) for the dorsal and ventral cranium in morphospace defined by the first two PC axes at the metamorphic stage. In deformation grids (C–F), blue circles represent each landmark and the set of landmarks describe mean cranial shape. Straight blue lines describe the shift in positions of landmarks from the mean shape to the target shape (C and E–positive part of the axis, D and F–negative part of the axis).

the expression of TH-alpha is characteristic of the larval period and is relatively constant from hatching to metamorphosis (*Terrien & Prunet, 2013*). TR-alpha receptors are widely distributed through all tissue and are present even before the organism has a functional thyroid gland (*Gilbert, 2010*). Also, thyroid hormone is among the major factors triggering tissue transformation, metabolism, growth, reproduction and metamorphosis in both anurans and urodeles. Therefore, experimental reduction of thyroid function with anti-thyroid substances better known as endocrine disruptors (substances like propylthiouracil, methimazole and thiourea) chemically blocks the synthesis of TH and causes a delay in larval development and metamorphosis (*Gobdon, Goldsmith & Charipper, 1943*; *Degitz et al., 2005*; *Thambirajah et al., 2019*). Lowering of the level of TH in the plasma caused by thiourea results in the failure of most TH-dependent events such as ossification of some cranial bones and skull remodeling, for which a high hormonal concentration is required (*Smirnov & Vassilieva, 2005*). On the other hand, experiments with exogenous applied TH to amphibian larvae led to precocious ossification of bones (*Terry, 1918*; *Fox & Irving, 1950*; *Dent, Kirby-Smith & Craig, 1955*; *Dent & Kirby-Smith, 1963*; *Kemp & Hoyt, 1965*; *Yeatman, 1967*; *Hanken & Hall, 1988*; *Smirnov & Vassilieva, 2003a*). Results obtained in

experiments on skull development during anuran metamorphosis revealed that exogenous applied TH initiates precocious ossification by accelerating growth and expanding the calcified matrix within already differentiated ossification centres (*Hanken & Hall, 1988*).

## Exposure to thiourea during the midlarval period

In the first experiment, larvae were exposed to two different concentrations of thiourea or were not treated during the midlarval period (from the stage when hind limb digits began to develop to the stage of fully developed limbs, digits and tail). The most notable result is that thiourea does not have such a dramatic effect on skull development during the midlarval period compared to skull development during metamorphosis. All analyzed cranial elements are present without any missing or resorbed ones, with a similar level of ossification in all individuals. Individuals from all analyzed groups had similar skull shape at stage 62, which indicates that larval cranial development proceeds similarly regardless of different TH levels during the midlarval period. This is in keeping with the results of previous studies treating thiourea as an endocrine disruptor that blocks the synthesis of TH.

According to *Smirnov & Vassilieva (2003a)*, early-appearing cranial ossification is TH-independent, but this independence does not mean that early-appearing bones are insensitive to TH, and even an undetectably low concentration of thyroid hormones in the blood was sufficient for regulation of their development. As we mentioned before, thiourea does not completely block activity of the thyroid gland, and a low level of thyroid hormones is probably still present in the blood. In comparison with early larval stages, during the midlarval period TH responsiveness increased and cranial bones became more TH dependent, which means that a higher hormone concentration in the blood plasma is required. For example, under conditions of different hormonal regimes, appearance and differentiation of early-appearing bones (vomer and palatopterygoid) proceed as in natural development during the early and midlarval period, but during metamorphosis their further development and rearrangements transpire differently, depending on the hormonal level (*Smirnov & Vassilieva, 2003a*).

## Exposure to thiourea during the late larval period and metamorphosis

During the late larval and metamorphosing periods, TH dependence greatly increases, and rising concentrations of TH circulating in the blood induce skull remodeling via bone formation, resorption and transformation (*Smirnov & Vassilieva, 2003a*). A study on the plethodontid salamander *Eurycea bislenata* using radioimmunoassay (RIA) revealed low levels of TH in pre-and post-metamorphic stages, with a peak hormone level at the metamorphic stage (*Alberch, Gale & Larsen, 1986*). Similar results were obtained for other urodeles (*Eagleson & McKeown, 1978*; *Larras-Regard, Taurog & Dorris, 1981*), but not for the axolotl *Ambystoma mexicanum*, which has an unusual plasma TH profile (a low level of circulating TH, low 5′- deiodinase activity and a low TH receptor number) and typically does not undergo a metamorphosis. In our experiment, individuals from the control group at the metamorphic stage display the normal skull development expected for the late

larval period and remodeling during metamorphosis. On the other hand, individuals from both thiourea-treated groups (low and high concentrations) retain the late larval skull without remodeling. Based on the results obtained in determination of ossification levels, it can be asserted that members of the control group have fully developed and ossified TH-dependent late-appearing bones (maxilla, nasal and prefrontal), unlike members of the treatment groups, which display serious retardation or even a lack of bone ossification. Complete disintegration of the palatopterygoid was present in the control group, unlike in thiourea-treated individuals where the palatopterygoid failed to disintegrate into the palatal and pterygoid portions. Larval organization of the vomer was retained in thiourea-treated larvae.

During ontogeny, the greatest changes of skull shape take place during metamorphosis. Larvae treated with two different thiourea concentrations had similar skull shape, significantly different from skull shape in members of the control group. All goitrogenic larvae retained the same skull shape as at the late larval stage or the stage of the onset of metamorphosis, even though they were at the same age as larvae of the control group. Differences between the control and thiourea-treated groups in the level of ossification, organization of the skull and its shape indicate that these processes are TH-dependent and that their development was halted in thiourea-treated larvae.

Our results confirmed that late-appearing bones (maxilla, nasal and prefrontal) are highly TH-dependent in *Triturus* newts. It is interesting that these three bones have different TH sensitivity in some other Urodela species. For example, all three bones are slightly TH-dependent in *Salamandrela keyserlingi*, moderately TH-dependent in *Ambystoma mexicanum*, and highly TH-dependent in *Lissotriton vulgaris* (*Smirnov, Vassilieva & Merkulova, 2011*). The obtained findings indicate the presence of interspecific variation in the TH-dependence of skull development in various species of Urodela. *Rose (1996)* proposed that in addition to changes in the level of the TH in the blood and different rate of thyroid gland activity, the evolution of Urodela was accompanied by changes in the TH dependence of some cranial bones. Also, previous studies showed that skull remodeling during metamorphosis is dependent on the applied dosage of TH, which points to interspecific and individual variation of bone TH sensitivity (*Rose, 1999*; *Rose, 2003*). However, in our study two applied thiourea concentrations gave essentially the same results in regard to the ossification level and skull remodeling, indicating that the difference in concentrations was insufficient to induce significant differences in the level of bone ossification and/or skull shape between individuals from the two thiourea-treated groups. Also, we examined histological sections of the thyroid glands of larvae that we included in this study (M. Ajdukovic, 2020, unpublished data). The thyroid gland of larvae treated during the late larval stage with both concentrations of thiourea showed notable hypertrophy and hyperplasia of thyroid follicles, which indicates that the thyroid glands were exhausted and TH synthesis was inhibited.

Our study confirms that in *Triturus* newts a deficiency in TH delays metamorphosis and induces retention of larval skull characteristics (the level of ossification of cranial elements and skull shape), the main feature of paedomorphic species. Paedomorphosis had significant effects on skull bone development in smooth (*Lissotriton* sp.) and alpine

(*Ichtyosaura* sp.) newts, with inter- and intraspecific variation in ossification levels of cranial elements. However, most adult paedomorphic individuals retain the same cranial skeletal organization as at the late larval stages (*Djorović & Kalezić, 2000*; *Ivanović et al., 2014*). Further studies of metamorphic and post-metamorphic development of newts treated with different concentrations of thiourea during ontogeny could clarify which TH levels are sufficient for metamorphosis, delay metamorphosis or completely block it and lead to paedomorphic adult individuals.

## CONCLUSION

Exposure to thiourea during the midlarval period has no very dramatic effect on the level of ossification of early-appearing cranial bones and does not affect the general skull shape in *Triturus* newts. Our results confirmed that early-appearing cranial ossification is TH-independent, but this independence does not mean that early-appearing bones are insensitive to TH. At the metamorphic stage, TH-dependent bones play a major role in establishment of skull shape via the level of cranial ossification, which is highly responsive to thiourea exposure during the late larval and metamorphic periods. Skull development during these periods is dependent on endogenous TH, and optimal TH values are required to maintain the normal rate of skeletal development and attain the appropriate final skull shape. Differences in skull shape due to the influence of thiourea as an endocrine disruptor at the late larval period and during metamorphosis between the control and thiourea-treated groups may cause different fitness of the groups. Future studies could explain how these differences are reflected in post-metamorphic development and what their main consequences are.

## ACKNOWLEDGEMENTS

We would like to thank many undergraduate students of the Faculty of Biology, University of Belgrade, for their technical assistance during the experiment.

### Funding

This work was supported by the Serbian Ministry of Education, Science and Technological Development of the Republic of Serbia (Grant numbers 451-03-9/2021-14/ 200007 and 451-03-9/2021-14/ 200178). The funders had no role in study design, data collection and analysis, decision to publish, or preparation of the manuscript.

### Grant Disclosures

The following grant information was disclosed by the authors:
Serbian Ministry of Education, Science and Technological Development of the Republic of Serbia: 451-03-9/2021-14/ 200007, 451-03-9/2021-14/ 200178.

### Competing Interests

The authors declare there are no competing interests.
## Author Contributions

- Maja Ajduković conceived and designed the experiments, performed the experiments, analyzed the data, prepared figures and/or tables, authored or reviewed drafts of the paper, and approved the final draft.
- Tijana Vučić and Milena Cvijanović performed the experiments, analyzed the data, prepared figures and/or tables, authored or reviewed drafts of the paper, and approved the final draft.

## Animal Ethics

The following information was supplied relating to ethical approvals (i.e., approving body and any reference numbers):

The experiment was approved by the Ethical Committee of the Institute for Biological Research ''Siniša Stanković'' (decision no. 02-07/19).

## Field Study Permissions

The following information was supplied relating to field study approvals (i.e., approving body and any reference numbers):

The Ministry of Energy, Development and Environmental Protection of the Republic of Serbia (permit no. 353-01-75/2014-08) and the Environmental Protection Agency of Montenegro approved field collections (permit no. UPI-328/4).

## Data Availability

The raw measurements are available in the Supplementary Files.

## Supplemental Information

Supplemental information for this article can be found online at http://dx.doi.org/10.7717/peerj.11535#supplemental-information.

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
