# Peer review of "Effects of thiourea on the skull of Triturus newts during ontogeny"

_PeerJ, doi:10.7717/peerj.11535_

## Round 0.1 · original submission · Major Revisions

I apologize for the delay. I have now received two reviews of your manuscript. Both reviewers identify major flaws in your study that should be fully considered. I am particularly concerned about your experimental design that seems not to be complete. Please clarify the experimental unit. Did you use individuals as the unity of measurement? Both reviewers asked a series of questions that should be considered to rewrite the manuscript. Some of your inferences regarding the nature of thyroid hormone control of cranial ossification during metamorphosis in amphibians seem not justified by your results. Your survey of the literature seems incomplete. Please, take all reviewer's suggestions into full consideration. I think they added valuable comments.


Reviewer 1 ·

Basic reporting

The manuscript is incomplete, insofar as not enough basic information about the study system is included to rigorously interpret the data or results. See specific comments below.

Syntax of figure legends is awkward in places.

Experimental design

Certain basic information that would explain and justify the experimental design is missing:

94 Likely, but are there any empirical studies that document the TH profile during metamorphosis in these species?

100-101 Why use interspecific hybrids and not larvae of one species or the other? Is the fact that they are hybrids make us expect that they would respond differently than larvae of each species?

112-114 Need to describe basic features of this staging table that are most pertinent to this study. For example, what are the key skeletal features of stages of greatest interest here—50, 62, and metamorphosis? These questions are not answered in Table S1.

119-120 What bones are normally present at these stages, and at what stages do bones form between them?

132-133 What bones are normally present at these stages, and at what stage(s) do bones form between them?

Validity of the findings

Certain inferences made regarding the nature of thyroid hormone control of cranial ossification during metamorphosis in amphibians are not justified by the results provided here. A more nuanced discussion of these results in terms of what they do--and don't tell us--is needed. There is a vast literature that describes thyroid gland activity and the nature of thyroid hormone action on target tissues. Little of that literature is cited here.

251-253 While one can safely conclude that there is differential TH-sensitivity between early- and late-forming bones, evidence for the claim that “cranial bones are not TH-dependent” is weak, despite claims made in published studies cited here. Neither the fact that early-forming bones may not be inducible by exogenous TH, nor that application of thiourea fails to prevent their formation, is conclusive. In the former instance, exogenous TH may have been applied before the bone-forming sites are competent to respond to it, and in the latter instance TH produced before application of thiourea may be sufficient to promote bone development after the application. These subtle relationships have a bearing on the way that results of the present study are interpreted.

258-265 While technically it may be correct to say that, for the experimental protocol used here, “thiourea does not have significant impact on skull development during midlarval period,” and that “larval cranial development proceeds similarly regardless of different TH level,” these claims need not mean that TH is not mediating the ossification of early-forming bones as is implied by the sentence that follows: “The thyroid gland begins its activity at the late larval period and has the greatest influence on late-appearing bones.” The pervasive effects of TH on metamorphosis at all stages, and I believe the past demonstration that the thyroid gland is active at if not before early stages, contradict the published claim cited here.

Additional comments

Minor corrections:

49 No hyphen in “triiodothyronine”

54 add comma after “metamorphosis”

145-146 Only the A in Alcian should be capitalized. The b in blue, a in alizarin, and r in red should all be lowercase.

369-371 This paper is cited out of chronological order.

384, 387 Need to correct spelling of second author’s name, here and in text.

436-437 Delete “for analyzed bones”

440-441 Delete “for analyzed bones.” palatopterygoid is misspelled.

Figure 2 Explain circles and squares. What does “of Inertia” mean?

Figure 3 Explain orientation of deformation grids. Anterior? Posterior? Dorsal? Ventral?

Reviewer 2 ·

Basic reporting

This manuscript describes experimental research into the influence of thyroid hormone deficiency on bone ossification and cranial morphology in developing Triturus newts. Thyroid hormones are well documented to have an important role in skeletal differentiation and morphogenesis in vertebrates and in amphibian metamorphosis, specifically. The influence of thyroid hormones on cranial bone appearance and ossification is well documented in Urodela (salamanders/newts); however, this study is the first study on skull shape in Urodela (based on information provided by the authors).

In order to evaluate the effect of thyroid hormone deficiency on skull development/shape, larval newts were exposed to a known disruptor of thyroid hormone synthesis, thiourea, which has previously been documented to alter thyroid hormone status, result in glandular hypertrophy, and processes dependent on thyroid hormones (e.g., metamorphosis in Anurans). The authors report on two experiments with which they were able to compare the effect of thyroid hormone deficiency (produced by exposure to two concentrations of thiourea) on cranial morphology during mid-larval and late larval development. Cleared and stained skull of skeletal whole-mounts were evaluated for presence and amount of ossification of skeletal elements with skull shape described/digitized using cranium landmarks.

1.1. This manuscript is well written with detailed descriptions of the work conducted and clarity in structure. To the best of my knowledge, the analyses conducted are appropriate. However, there are aspects of the reporting, experimental design, and findings that need clarification and/or expansion in order to provide support for the conclusions made, as described below.

1.2. The introduction provides a good overview of the importance of THs in vertebrate development; however, for the context of thyroid disruption, it is important to include details on the mode of action of thiourea and documented effects on circulating thyroid hormones, beyond the generalities described here.

1.2.1. In the Intro, I recommended adding information on the mechanism by which thiourea reduces THs – likely through inhibition of thyroperioxidase (TPO) enzyme activity, as suggested by the use of thiourea-type compounds included as reference chemicals for TPO in Wener et al. 2016
[Wegner S, Browne P, Dix D. Identifying reference chemicals for thyroid bioactivity screening. Reprod Toxicol. 2016 Oct;65:402-413. doi: 10.1016/j.reprotox.2016.08.016]

1.2.2. Although thiourea is well documented as a model thyroid disruptor, details on documented reductions in circulating thyroid hormones from previous publications should be added to support the premise of this study. If possible, include reference to studies that documented reductions in T3 and/or T3 in amphibians – either in the Intro or the Discussion.

1.3. It is unclear why interspecific crosses were used for this study. Would the expectation of effects be the same in both of these species?

1.4. The figures and tables are relevant, well-described, and support the text of this manuscript.

1.5. The raw data are provided for the ossification scores from both experiments. These data are easy to understand, and my only recommendations are to correct the typo on the ‘concentration’ heading and perhaps indicate the compound (‘thiourea’) in either the header or the entries for this field. I am not sure if it is possible to provide the raw data for the digitized cranium landmarks – but this information could be of interest to some.

Experimental design

2.1. It is not clear what the experimental unit is in these studies – the individual (at with scoring and landmarks were done) or the container (Line 129 indicates that there were 5 larvae per container). Please clarify the experimental unit, how many containers, and which were used for the statistical analysis. The sample sizes reported suggest that there were only 2-4 containers per treatment each with 5 larvae.

2.2. It would be beneficial to expand upon the selection of concentrations of thiourea (Lines 123-124). I appreciate the citation for the concentration selection, but suggest expanding on the effects these concentrations had in the citation used (and possibly add other citations). Also, was thiourea dissolved in the same tap water as used for the controls?

2.3. Please clarify which data were used for the PCA for variation in skull shape. Scores on presence and ossification level (described on Lines 163-165?) Cranium landmarks?

2.4. This research was conducted under ethic standards following national permits and institutional approval, and in compliance with the European Directive 2010/63/EU. The described exposures meet the standards in the field, with some additional information that should be reported: details on exposure (described below in 3.2) and method of euthanasia. The research questions could only be addressed with these experiments, and the authors used the minimum number of organisms (perhaps fewer than would be recommended, as described above in 2.1).

Validity of the findings

3.1. The design of these experiments provided a good basis for addressing the question of effect of TH deficiency on skull development/shape, including controls and 2 treatment levels at two different developmental stages. However, it is challenging when development and/or metamorphosis is delayed and the treatments cannot include organisms that are staged-matched with the controls. The authors have recognized this challenge, but I recommend that further descriptions or information are needed to determine whether the differences in the study with late larval/metamorphic newts are due to actual changes in skull development/shape due to TH deficiency, or are the exposed organisms just delayed in development and have skull shapes similar to an earlier stage. If possible, perhaps it would be informative to compare the ossification scores and shape metrics of treated organisms to this same information from control organisms at multiple earlier stages.

3.2. To improve confidence in the results reported, I suggest adding other details on the exposure. Specifically:
3.2.1. What was the length of exposure for each of the studies? Did they differ greatly?
3.2.2. What as the density per container by mass? And how did that differ between the two studies?
3.2.3. What were the conditions in the containers? (Dissolved oxygen, pH, temperature, etc).
3.2.4. Was there much mortality observed in either study? What about other growth effects?

Additional comments

4.1. Line 54: I suggest revising “secretion of TH” to more specific description of “TH synthesis and release to bloodstream increases” or “circulating TH increases”.

4.2. Lines 54-56: Consider adding additional citations to support the pattern of TH through the metamorphic process. There are multiple studies with amphibians that would supplement the 2 citations already included, such as: Regard et al. 1978, Larras-Regard et al. 1981, Sternberg et al. 2011 (and many others).

Regard E, Taurog A, Nakashima T. Plasma thyroxine and triiodothyronine levels in spontaneously metamorphosing Rana catesbeiana tadpoles and in adult anuran amphibia. Endocrinology. 1978 Mar;102(3):674-84. doi: 10.1210/endo-102-3-674.

Larras-Regard E, Taurog A, Dorris M. Plasma T4 and T3 levels in Ambystoma tigrinum at various stages of metamorphosis. Gen Comp Endocrinol. 1981 Apr;43(4):443-50. doi: 10.1016/0016-6480(81)90228-8.

Sternberg RM, Thoemke KR, Korte JJ, Moen SM, Olson JM, Korte L, Tietge JE, Degitz SJ Jr. Control of pituitary thyroid-stimulating hormone synthesis and secretion by thyroid hormones during Xenopus metamorphosis. Gen Comp Endocrinol. 2011 Sep 15;173(3):428-37. doi: 10.1016/j.ygcen.2011.06.020.

4.3. Lines 80-81: Were these studies on cranial osteology conducted in tadpoles (as mentioned in the first ½ of the sentence) or in other amphibian groups?

4.4. Lines: 82-85: For the more general audience associated with this journal, it would be helpful to provide context for what are included in the Order: Urodela (salamanders/newts).

4.5. Line 124: I suggest moving the details on the action of thiourea to the Intro (instead of including here in the Methods) – it would fit well with the sentence on lines 71-73. I also recommend adding the CASRN and/or Product ID from the thiourea obtained from Sigma, along with the reported purity of the lot received.

4.6. Line 131: “They were sampled.” Euthanized with ??, preserved whole? Any tissues or blood collected?

---

## Round 0.2 · Minor Revisions

I am sorry for the delay in making my decision. Thank you for considering the reviewer's suggestion. Some issues persist, however. Please make a new paragraph in the Introduction section with the sentences devoted to present the Urodela order. You have explained in the rebuttal letter your reasons to make your experiments with a hybrid taxon, this explanation should be added to the manuscript.

Other points were detailed in the response and not explained in the text. Please, consider the new suggestions of our reviewers and make your text more detailed. Early on in your experimental setting, the number of specimens used should be stated. Please, incorporate to the epigraph of your Figure 3 the explanation given in the rebuttal letter: "Blue circles represent each landmark and set of landmarks describe mean cranial shape. Straight blue lines describe the shift of positions of landmarks from the mean shape to the target shape (first and third graph – positive part of the axis, second and fourth graph – negative part of the axis; it is marked by an arrow on the left side of deformation grids)." You have mitigated your interpretations as our reviewer suggested, however, the conclusions are as strong as before, please rewrite them.

Reviewer 1 ·

Basic reporting

English grammar and syntax is still awkward in places in the text. Someone in the editorial process will need to fix.

Specific errors (line numbers in parentheses):
76 Delete comma after “agent”.
86 Change “as” to “a”.
118 treated is misspelled (“tread”).
330 Eagleson is misspelled (“Eagieson”) here and in Literature Cited.
402 Add “of the United States of America” after “Sciences”.
417 Capitalize first a of “alcian”.
426 Eagleson.
526 Insert space after “Lithobates”.

thiourea is misspelled (“tiourea”) in one of the Supplemental files (peerj-55888-skull_shape_raw_coordinate_stage_62.xlsx”).

Experimental design

In their response to reviewers, the authors provide a helpful explanation and justification of the use of hybrid larvae, but little of that text has made it into the revised manuscript (lines
123-127). I suggest inserting more of the explanation and justification into the manuscript to allay readers' concerns.

Validity of the findings

Problematic claims from the original draft have been removed or modified appropriately.

Reviewer 2 ·

Basic reporting

The authors have thoroughly responded to the reviewers’ comments and recommendations. The additional details added to the text and supplementary materials improve the manuscript and interpretation of the results. However, some of the details provided in the response letter were not added to the text. I recommended adding a few of these details (listed below) in order to support the context and interpretation of these studies.

1. The authors provided detailed responses and added text for much of the basic reporting. In the revised text, I suggest adding a sentence clarifying why hybrids were used. The descriptions of the reason are well described in the responses to both reviewers but still is not well described in the text. This would fit well added to the sentence on Lines 126-127 or added to the last paragraph of the Introduction.

2. I greatly appreciate the additional information the authors added on the importance of TPO for biosynthesis of thyroid hormones. However, I suggest removing the sentence that was added on Lines 84-86 and keeping the focus on TPO. It is confusing to the reader to include inhibition of thyroidal iodine uptake (driven by the sodium-iodide symporter) and conversion of T4 to T3 (driven by iodothyronine deiodinases).

Experimental design

3. The authors addressed the questions regarding experimental unit and justified the minimal number of organisms used in these studies.

Validity of the findings

4. The responses clearly describe that there were no differences in the conditions in the containers (DO, pH, temperature) and no difference in mortality or growth across treatments. These provide confidence in the quality of the experiments, and, thus, I suggest including these observations in the text.

5. In the response to Reviewer 2 Question 3.1, the authors describe histological analysis of the thyroid glands of this study. The results described in the response letter would provide great confidence in reduction of thyroid hormone synthesis in the thiourea treated organisms. If possible, I recommend added these qualitative results to the discussion to support the statements about thyroid hormone levels.

Additional comments

Minor suggestions:
a. Line 73: Suggest moving source and purity of Thiourea back to the methods, but keeping the CAS Number in the Intro where Thiourea is first discussed.
b. Line 118: Change ‘tread’ to ‘treated’
c. Line 157: Remove the word ‘about’
d. Line 170: Fix typo in ‘Supplementary’
e. Line 174: Change ‘leading’ to ‘leads’

---

## Round 0.3 · accepted · Accept

Thank you for your consideration of all suggestions made to improve your study.